# Dynamic Regulation of *brsk2* in the Social and Motor Development of Zebrafish: A Developmental Behavior Analysis

**DOI:** 10.3390/ijms242216506

**Published:** 2023-11-19

**Authors:** Jingxin Deng, Chunxue Liu, Meixin Hu, Chunchun Hu, Jia Lin, Qiang Li, Xiu Xu

**Affiliations:** 1Division of Child Health Care, Children’s Hospital of Fudan University, National Children’s Medical Center, 399 Wanyuan Road, Shanghai 201102, China; 21111240005@m.fudan.edu.cn (J.D.); 22111240011@m.fudan.edu.cn (M.H.); 14211240007@fudan.edu.cn (C.H.); 2Center for Translational Medicine, Institute of Pediatrics, Shanghai Key Laboratory of Birth Defect, Children’s Hospital of Fudan University, National Children’s Medical Center, 399 Wanyuan Road, Shanghai 201102, China; linjiaalyssa@163.com (J.L.); liq@fudan.edu.cn (Q.L.)

**Keywords:** *BRSK2*, zebrafish, motor, social, development, dynamic regulation

## Abstract

Both social and motor development play an essential role in an individual’s physical, psychological, and social well-being. It is essential to conduct a dynamic analysis at multiple time points during the developmental process as it helps us better understand and evaluate the trajectory and changes in individual development. Recently, some studies found that mutations in the *BRSK2* gene may contribute to motor impairments, delays in achieving motor milestones, and deficits in social behavior and communication skills in patients. However, little is known about the dynamic analysis of social and motor development at multiple time points during the development of the *brsk2* gene. We generated a novel *brsk2*-deficient (*brsk2ab*^−/−^) zebrafish model through CRISPR/Cas9 editing and conducted comprehensive morphological and neurobehavioral evaluations, including that of locomotor behaviors, social behaviors, and anxiety behaviors from the larval to adult stages of development. Compared to wild-type zebrafish, *brsk2ab*^−/−^ zebrafish exhibited a catch-up growth pattern of body length and gradually improved locomotor activities during the developmental process. In contrast, multimodal behavior tests showed that the *brsk2ab*^−/−^ zebrafish displayed escalating social deficiency and anxiety-like behaviors over time. We reported for the first time that the *brsk2* gene had dynamic regulatory effects on motor and social development. It helps us understand developmental trends, capture changes, facilitate early interventions, and provide personalized support and development opportunities for individuals.

## 1. Introduction

Motor and social development are both crucial aspects of an organism’s overall development and well-being. Motor and social development are interconnected and deeply intertwined. Movement skills enable individuals to actively engage in social interactions, while social interactions provide the context for the development and refinement of movement abilities [1,2]. Both aspects are essential for navigating and thriving in physical and social environments. It is important to recognize that motor and social development are intricate, influenced by genetics, the environment, and individual differences, resulting in unique developmental paths. Genetic and environmental factors jointly regulate these developments, encompassing mechanisms like that of genetic and environmental regulation, as well as developmental processes [3,4,5]. Specific genes such as the *SHANK3* [6,7,8,9] and *BRSK2* [6,10,11] genes impact motor and social development. These genes oversee the creation and function of neural circuits, muscle growth, and neurotransmitter production tied to motor control and social behavior. Variations in these genes can sway the course of motor and social development. Critical periods often host motor and social development, capitalizing on the brain’s heightened sensitivity to environmental cues. During these phases, genetic and environmental factors wield a pronounced influence over shaping these facets. The brain exhibits plasticity—the capacity to adapt its structure and function through experiences. As individuals engage in motor and social activities, neural connections are fortified or refined, inducing adaptive changes in both motor and social skills.

The *BRSK2* gene encodes the protein BRSK2 (Brain-specific serine/threonine-protein kinase 2), a member of the serine/threonine kinase family [12]. *BRSK2* is predominantly expressed in the human brain, especially in regions linked to neuronal development [13], while in mice, *Brsk2* is limited to the nervous system [12,14]. Although the function of the *BRSK2* gene is still under investigation, earlier research has indicated its involvement in neuronal development and plasticity, cell polarity and migration, the formation of presynaptic vesicles, axonal development, neuronal polarization, and neurological disorders [12,14,15,16,17,18,19,20]. *BRSK2* deficiency has been linked to numerous neurodevelopmental and neurological disorders, including autism spectrum disorder (ASD), developmental delays, and/or intellectual disability [10,11].

*BRSK2* has been implicated in various aspects of motor and social development. Although the precise mechanisms are not fully understood, studies have shed light on the role of *BRSK2* in these processes. Mutations in the *BRSK2* gene may contribute to motor impairments and delays in achieving motor milestones in patients [10,11]. Similarly, *brsk2b* knockout zebrafish displayed reduced swimming patterns [6]. Disruptions in *BRSK2* function have been linked to possible difficulties with fine motor skills [10,11]. Furthermore, research indicates that patients with mutations in the *BRSK2* gene exhibited deficits in social behavior and communication skills, which are pivotal aspects of human interaction that significantly influence different facets of life [10,11]. This implies that *BRSK2* might be involved in the neural circuits and molecular processes that underlie social behavior and social cognition.

The zebrafish (Danio rerio) is widely employed as a model organism in scientific research owing to its genetic resemblance to humans and its transparent embryos, facilitating the convenient observation of its internal structures [21]. Zebrafish development is governed by an array of genetic and environmental factors, which regulate essential aspects such as social behavior and motor skills [21,22,23,24,25].

In conclusion, comprehending the regulation of *BRSK2* expression is crucial in uncovering its involvement in diverse biological processes, such as social behavior and motor development. Investigating the modulation of *BRSK2* expression offers insights into the intricate molecular mechanisms driving these behaviors and their potential disruption in neurodevelopmental disorders. However, limited research has explored the regulatory functions of *BRSK2* in motor and social development processes. In this study, we employ the zebrafish model to examine the dynamic regulation of the *brsk2* gene in relation to motor and social interaction across various developmental stages.

## 2. Results

### 2.1. Temporal Expression of brsk2 in Zebrafish

To differentiate the developmental profiles of *brsk2a* and *brsk2b* in zebrafish, we conducted transcript-specific qRT-PCR from wide-type (WT) zebrafish at eight different developmental stages: 1, 2, 3, 5, 7, and 10 days post-fertilization (dpf), 1 month post-fertilization (mpf), and 3 mpf.

At early ages (prior to 1 dpf), both *brsk2a* and *brsk2b* exhibited minimal expression levels. However, their expression levels surged after 1 dpf and continued to rise throughout the zebrafish larvae’s development (Figure 1 and Appendix A). Generally, *brsk2b* exhibited higher expression levels compared to *brsk2a* across various developmental stages. Both *brsk2a* and *brsk2b* demonstrated a gradual increase in expression up to 5–7 dpf. A peak in expression occurred around 7 dpf for *brsk2a* and 5 dpf for *brsk2b.* Remarkably, both *brsk2a* and *brsk2b* demonstrated a subsequent decline in expression post 7 dpf. Additionally, during the juvenile and adulthood stages, both *brsk2a* and *brsk2b* displayed a second peak of increased expression.

### 2.2. Generation of brsk2a^−/−^ and brsk2ab^−/−^ Zebrafish

To investigate the regulatory role of *brsk2* during development, we employed CRISPR/Cas9 mutagenesis to create a loss-of-function *brsk2* mutant. Specifically, the *brsk2b*^−/−^ strain exhibited a 2-bp insertion, resulting in a premature stop codon and truncated protein (p.K117Rfs*27), which has been described in our previous study [6]. Meanwhile, the *brsk2a*^−/−^ line featured a 23-base pair deletion (p.V284Qfs*5), causing a frameshift mutation and yielding a 288-amino acid truncated protein (Figure 2A,B). By crossing *brsk2b*^−/−^ and *brsk2a*^−/−^, we established the *brsk2ab*^−/−^ line, subsequently identifying individuals through genotyping. The qRT-PCR verified a marked reduction in *brsk2a* and *brsk2b* mRNA expression levels in *brsk2ab*^−/−^ zebrafish compared to the WT counterparts (Figure 2C).

### 2.3. A Catch-Up Growth Pattern in brsk2-Deficient Zebrafish

The body length of zebrafish serves as an indicator of their growth, developmental status, health, and adaptability to the environment. Body length measurements were taken at different developmental stages (e.g., 2 dpf, 3 dpf, 5 dfp, 2.5 mpf, and 3 mpf; Figure 3). During the larval stage, *brsk2ab*^−/−^ zebrafish displayed notably shorter body lengths than their WT counterparts. At 2 dpf, the *brsk2ab*^−/−^ zebrafish (3.623 ± 0.09 mm, *p* < 0.0001) exhibited significantly reduced body length compared to the WT (3.760 ± 0.09 mm). However, accelerated growth occurred subsequently, resulting in a gradual decrease in the disparity between the two groups. In the late juvenile (2.5 mpf, WT vs. *brsk2ab*^−/−^: 21.19 ± 1.58 mm vs. 20.90 ± 2.31 mm, *p* = 0.646) and adult (4 mpf, WT vs. *brsk2ab*^−/−^: 27.57 ± 1.66 mm vs. 28.04 ± 1.53 mm; *p* = 0.425) stages, there was no significant difference in body length between the *brsk2ab*^−/−^ zebrafish and the WT zebrafish.

### 2.4. Gradually Improved Locomotor Activities during the Developmental Process in brsk2-Deficient Zebrafish

The body length of zebrafish is linked to the development of their motor function. As zebrafish grow and develop, their body size and shape can influence their motor skills. Hence, we evaluated the locomotor behaviors of the zebrafish across various developmental stages (Figure 4 and Appendix A). The *brsk2ab*^−/−^ zebrafish remained viable and fertile into their adulthood. Compared to WT zebrafish (3.342 ± 1.63 mm/s), the *brsk2ab*^−/−^ zebrafish (2.229 ± 1.18 mm/s, *p* = 0.0002) exhibited significantly reduced swimming velocity at 5 dpf. Subsequently, the motor function of the *brsk2ab*^−/−^ zebrafish gradually improved, leading to a progressive decrease in the discrepancies between the two groups. By 14 dpf, there was no significant disparity in swimming speed between the *brsk2ab*^−/−^ zebrafish (3.826 ± 1.18 mm/s, *p* = 0.2753) and the WT fish (4.046 ± 1.50 mm/s), and this undifferentiated state persisted throughout the juvenile (2 mpf) and adulthood (3 mpf) stages. The gradual refinement of their physical development may contribute to their motor function improvement. In summary, greater body length generally correlated with enhanced swimming ability and motor coordination, suggesting an interconnectedness between body development and motor function during the growth process.

### 2.5. Influence of brsk2 on SOCIAL Development Strengthens as Development Progresses

Neural development and remodeling play a pivotal role in the acquisition of social skills. To assess the social capabilities of the zebrafish across various time points, we employed multimodal social experiments (Figure 5A–C). At 1 mpf, a subtle, substantial distinction in social deficiency was observed between the *brsk2ab*^−/−^ zebrafish (*p* = 0.0298, Figure 5D) and the WT zebrafish. However, from 1.5 mpf onward, discernible differences began to surface between the two groups, with the *brsk2ab*^−/−^ zebrafish (*p* = 0.0005) displaying social impairments and significantly underperforming in social skills compared to the WT group (Figure 5E). At adulthood, the disparities between the two groups became more evident. Social preference was assessed by measuring conspecific proximity. The WT zebrafish maintain closer proximity with conspecific group members on the right side, as shown in Figure 5B,F. In contrast, the *brsk2ab*^−/−^ zebrafish exhibited a decreased frequency and duration of conspecific proximity at 3 mpf (distance, *p* = 0.0022; time, *p* = 0.0029). At 4 mpf (Figure 5H), the *brsk2ab*^−/−^ zebrafish displayed a comparable pattern of social deficiency behaviors (distance, *p* = 0.0072; time, *p* = 0.0044).

Additionally, the shoaling test was further gauged through schools (Figure 5C). By 3 mpf, the shoaling test indicated that the adult WT zebrafish predominantly spent their swimming time in tightly-knit schools (*p* < 0.0001, Figure 5G). Conversely, the *brsk2ab*^−/−^ zebrafish formed looser and larger schools, with more frequent deviations, resulting in a greater average inter-fish distance compared to the WT zebrafish. This pronounced social deficit was particularly notable among the *brsk2ab*^−/−^ zebrafish group at 4 mpf (*p* < 0.0001, Figure 5I).

In conclusion, social development and neural growth are intricately linked and reciprocally impactful. Healthy neural development promotes healthy social development, whereas abnormal neural development could harm social abilities.

### 2.6. brsk2ab^−/−^ Zebrafish Displayed Escalating Anxiety-like Behaviors over Time

An intricate connection exists between anxiety behaviors and the development of the nervous system as well as social behaviors. Neurodevelopmental factors can contribute to the emergence of anxiety-related behaviors, and social experiences also influence an individual’s anxiety levels. Classic open field experiments are employed to assess anxiety-like behavior in zebrafish. At 2 mpf, the *brsk2ab*^−/−^ zebrafish did not exhibit substantial anxiety-like behaviors in comparison to the WT group (distance, *p* = 0.1622; time, *p* = 0.2225; Figure 6A,B). Nevertheless, at 3 mpf, the *brsk2ab*^−/−^ zebrafish displayed significantly more pronounced anxiety-like behaviors compared to the control group (distance, *p* < 0.0001; time, *p* = 0.0038; Figure 6C).

## 3. Discussion

In this study, we generated the first *brsk2*-deficient zebrafish line and investigated the dynamic regulation of the *brsk2* gene on the development of motor and social interactions throughout the zebrafish life cycle. Morphologically, the *brsk2ab*^−/−^ zebrafish displayed a notable reduction in body length during the larval stage, followed by accelerated growth during the juvenile period, and eventually exhibited normal body length and morphology during adulthood. In terms of motor development, the *brsk2ab*^−/−^ zebrafish demonstrated a decline in motor function in the early stages of life, followed by a rapid pattern of catching up. Motor function remained normal until early adolescence. In terms of social development, there are no significant abnormalities in social function during the early juvenile stage for the *brsk2ab*^−/−^ zebrafish. However, social impairments have become increasingly apparent from the mid-juvenile stage onward in the *brsk2ab*^−/−^ zebrafish. Anxiety behaviors also exhibit a similar pattern. Understanding the role of *brsk2* in motor performance and social interaction during various developmental processes provides valuable insights into the dynamic nature of these influences.

Dynamic analysis at multiple time points during development provides a comprehensive understanding of the complex interplay between genes, the environment, and individual outcomes. It enables us to capture the temporal dynamics, cumulative effects, and critical periods that shape sports performance, social interaction skills, and other aspects of development. Firstly, development is a continuous process, and different changes may occur at different time points. By performing a dynamic analysis at various time points throughout development, we can detect patterns and trends in these changes, enhancing our comprehension of the dynamic nature of development. Secondly, it is crucial to recognize critical periods, during which certain developmental milestones and significant changes occur. By analyzing development at various time points, critical periods can be identified to understand their importance in the development of specific abilities and behaviors. Thirdly, analyzing developmental data dynamically at different time points allows for the inference of trends and potential outcomes for future development. This is essential for formulating intervention strategies and providing support for development. For example, in this study, the expression levels of *brsk2a* and *brsk2b* increased in the first 5–7 days and then showed a slight decreasing trend, but they subsequently exhibited upward trend towards the juvenile and adulthood stages. The specific mechanisms behind this phenomenon were unclear. Here are some possible explanations: During early embryonic stages, various organs and tissues are rapidly developing and differentiating. *brsk2* may be involved in specific growth signaling pathways or regulatory mechanisms, including the structural and functional maturation of synapses [15], leading to an increase in its expression during this stage. As organ development completes and enters the stage of functional maturity, the expression level of *brsk2* may gradually decrease. During individual development, different regulatory mechanisms and molecular signaling pathways may undergo changes. These changes may result in a re-increase in the expression level of *brsk2* in the juvenile and adulthood stages, possibly due to new regulatory factors or signaling pathways beginning to influence its expression. In summary, conducting a dynamic analysis at multiple time points throughout development enables a more comprehensive and in-depth comprehension of the dynamic nature of development. It helps identify critical periods, shed light on individual differences, and predict future progress.

The motor function of *brsk2ab*^−/−^ zebrafish improved gradually throughout the study. The improvement in movement may be linked to the gradual normalization of body shape and length in these fish. The zebrafish is a widely researched model organism in the fields of neural function and motor development [21,22,23,24,25,26]. The enhancement in body shape and length may suggest typical growth and development, which could be linked to the progress in motor function. Little was previously known regarding the effect of a *brsk2* gene deficiency on motor function. It was confirmed that *brsk2* was necessary for the maturation of synapse structure and performance [15]. *brsk2* managed synapse maturity at the neuromuscular junction and undertook successive functions in the creation of sensory synapses on motor neurons [15]. So far, there have been limited reports of patients with pathogenic mutations in the *BRSK2* gene [6,10,11]. The majority of these cases involved children or adolescents, with 75% displaying gross motor developmental delays. Only one patient, aged 19, did not exhibit a gross motor delay. However, data pertaining to adult patients is lacking so far.

Motor skills are crucial for children’s development, as they make up a significant aspect of behavioral growth. It is important to note that motor skills are an integral part of overall development. These skills stimulate the changes involved in behavioral development by affecting and shaping behavior. In addition, motor development ensures that behavior can be functional and flexible enough to cope with a variable body in a variable world, excluding subjective evaluations unless explicitly stated [3]. Motor movement is generated by the activity of neuronal circuits, which collect and integrate information, ultimately leading to precisely timed skeletal muscle contractions [27]. Developmental and genetic programs instruct specific connectivity patterns, binding diverse neuronal subpopulations into motor circuit modules at every level of hierarchy [27]. 

Social behavior and communication skills play a vital role in human interaction and are key to successful functioning in many areas of life. They are critical for interpersonal connections, psychological well-being, and effective functioning within different social contexts. These abilities foster comprehension, collaboration, and genuine interactions, resulting in improved personal and professional relationships. In this study, we discovered a significant association between the loss-of-function of *brsk2* and social dysfunction, which worsens with age. Among the currently reported patients, 86% were diagnosed with ASD [6,10,11]. Feliciano et al. provided strong evidence that *BRSK2* is a high-confidence ASD risk gene [11]. Especially, the most prominent feature of ASD involves persistent deficits in interaction and social communication, including the social reciprocity and verbal communication behaviors used in social interactions and in the ability to develop and maintain social relationships [5]. Language and social skills are essential for individuals’ intrapersonal and interpersonal functioning and quality of life [4]. Over the last decade, imaging studies have provided crucial insights into the cortical and subcortical networks responsible for various social functions and their intricate integration [4]. Structural and functional imaging studies have found that the social affective network comprises the default mode, frontal, parietal, somatosensory, and temporal cortex, along with the subcortical thalamus, cerebellum, caudate, and pallidum. This integrated network subserves a wide range of social functions [4].

Similar to the pattern of social impairments, we discovered that the anxiety behavior of the *brsk2ab*^−/−^ zebrafish became more pronounced with age as the expression level of *brsk2* increased. A possible vicious cycle exists between anxiety and social behavior. Anxiety may prompt the avoidance of social behavior, which in turn amplifies anxiety. This cycle could lead to individuals slowly withdrawing and isolating themselves, making it challenging to create and sustain healthy social relationships. The basolateral amygdala complex (BLA) is a significant neural hub for the modulation of anxiety-like behaviors [28,29,30] and social behaviors [31,32]. The medial prefrontal cortex (mPFC) shared reciprocal projections with the BLA [33,34], exhibiting profound alterations in a wide range of anxiety and social disorders [35]. Felix-Ortiz et al. demonstrated that the BLA-mPFC projection plays a bidirectional role in modulating anxiety-related and social behaviors. Activating this pathway increases anxiety-like behavior and reduces social interaction, while inhibiting it reduces anxiety-like behavior and increases social behavior [36].

In conclusion, understanding the dynamic regulation of *brsk2* in zebrafish development can provide insights into the genetic mechanisms underlying social behavior and motor skills. By studying the role of *brsk2* in zebrafish, researchers can gain valuable knowledge about the molecular pathways involved in these processes, which may have implications for understanding similar processes in humans and other vertebrates.

## 4. Materials and Methods

### 4.1. Zebrafish Maintenance and Husbandry

The wild-type Tu strain was used in this study. All zebrafish were raised and maintained under the standard laboratory conditions of 28.5 °C under a 14-h light/10-h dark photoperiod in a recirculation system according to standard zebrafish breeding protocols [37]. For breeding, adult zebrafish were placed in tanks overnight with a sex ratio of 1:1 and were separated by a transparent barrier that was removed on the following morning. Freshly fertilized eggs were collected within 0.5 h of spawning, immersed in the methylene blue solution, and maintained in an incubator at 28.5 °C until 5 dpf. Larvae aged 5–12 dpf were nourished with paramecium and subsequently transferred to a recirculating water system at 28.5 °C. Water supplied to the system was filtered through reverse osmosis (pH 7.0–7.5). All animal experimental procedures were in compliance with local and international regulations, and approved by the institutional animal care committee of the Children’s Hospital of Fudan University.

### 4.2. Generation of brsk2ab Defect Zebrafish Line

The zebrafish *brsk2a* and *brsk2b* genes, along with their exon/intron boundaries, were identified in the NCBI database (gene ID: *brsk2a*, NC_007136.7; *brsk2b*, NC_007118.7). The Mutant zebrafish for *brsk2a* and *brsk2b* were generated using a CRISPR/cas9 system as previously reported [38,39]. The single guide RNA (sgRNA) of *brsk2a* was designed to target exon 10 (target site:5′-GGAGCAGCCGGTACCCAGGA-3′), while that of *brsk2b* [6] was located on the exon 4 (target site: 5′-GGGCAGGTTAACACCCAAAG-3′) and synthesized using the in vitro transcription kit (MAXIscriptTM T7 kit AM1314M, Invitrogen, Carlsbad, CA, USA). The fertilized WT zebrafish embryos (F0) at one-cell stage were co-injected with 600 pg cas9 Nuclease (EnGenTM spy cas9 NLS #M0646, New England Biolabs, Ipswich, MA, USA) and 150 pg gRNA. The genotyping protocol was detailed in a previous publication [6]. The F0 adult zebrafish were out-crossed with the WT zebrafish for at least three generations and genotyped at each generation to obtain F4 heterozygous zebrafish. The homozygous *brsk2a*^−/−^ or *brsk2b*^−/−^ zebrafish were obtained through genotype identification after in-crossing the F4 heterozygous adult zebrafish. The *brsk2ab*^−/−^ homozygous line was obtained by crossing homozygous *brsk2a*^−/−^ and *brsk2b*^−/−^ and subsequently genotyping (Appendix A). The genotyping PCR amplification conditions were as follows: 95 °C, 4 min; 35 cycles of 95 °C, 30 s; 58 °C, 30 s; 72 °C, 45 s; 72 °C, 7 min. The PCR primers for genotyping were listed in Appendix A.

### 4.3. Morphological Evaluation of brsk2ab^−/−^ Zebrafish

We measured the body length of the *brsk2ab*^−/−^ zebrafish at developmental stages of 2 dpf, 3 dpf, 5 dpf, 2.5 mpf, and 4 mpf. The larvae were anesthetized with tricaine (40 mg/L concentration) and lateral and dorsal images were captured using a stereo light microscope (Leica M205 FA, Wetzlar, Germany). The measurement of body length involved determining the distance from the head to the end of the tail fin. All acquired images were analyzed using Fiji (ImageJ) software (https://imagej.net/ij/ accessed on 12 July 2023).

### 4.4. Quantitative Real-Time Polymerase Chain Reaction (qRT-PCR)

The brain tissues from the zebrafish were collected at 4 mpf and utilized for qRT-PCR assay. The total RNA extraction was carried out using the RNAiso plus kit (No.9108, Takara Biomedical Technology, Shiga, Japan) following the manufacturer’s protocol. The reverse transcription of one microgram of RNA template was performed using the PrimeScript™ RT reagent Kit with gDNA Eraser (RR047, Takara Biomedical Technology, Shiga, Japan) according to the instructions. The resulting cDNA solution was diluted fivefold with RNase-free water, and 2 μL of the cDNA template was added to a 20 μL qRT-PCR system using a TB Green Premix Ex Taq II (Tli RNaseH Plus) (RR820A, Takara Biomedical Technology, Shiga, Japan) on a LightCycler R 480 apparatus (Roche, Basel, Switzerland). Reaction conditions were set according to the manufacturer’s instructions: 30 s of pre-denaturation (95 °C), 5 s of denaturation (95 °C), 30 s of annealing (60 °C), repeated for 40 cycles. The primer sequences for qRT-PCR can be found in Appendix A. The relative levels of target mRNA normalized to β-actin were calculated using the 2-∆∆Ct method. Each experiment was conducted three times.

### 4.5. Behavioral Test

#### 4.5.1. Larval Activity Test

Larval locomotor activity was assessed at 5 dpf and 14 dpf using a ViewPoint setup (ViewPoint Life Sciences, Lyon, France) coupled with an automated recording system, and evaluated using a zebralab software (https://www.viewpoint.com, accessed on 12 July 2023) [40,41,42]. Twenty-four-well plates were employed to accommodate a single larva per well, ensuring ample water volume for swimming. Following a 5-min acclimatization period, each test video was recorded for 10 min, with data on time and distance collected at 30 s intervals for each well. Locomotor activity was quantified as the mean distance moved per 30 s, and the average speed was subsequently calculated.

#### 4.5.2. Open Field Test

The open field test was performed to measure the locomotor activity and anxiety-like behavior of the adult zebrafish at 2 mpf and 3 mpf. The tests were conducted between 10:00 and 16:00 each day, with the test order alternating between different genotype groups to minimize potential circadian rhythm effects. The zebrafish were allowed to swim individually in a 30 cm × 30 cm × 30 cm opaque tank for 15 min of video recording (5 min for the adaptation phase and 10 min for the experiment phase). The time and distance data were collected every 30 s from the transformational visual route of the fish trajectory, utilizing Zebralab software (ViewPoint Life Sciences, Lyon, France). The locomotor movement was evaluated by calculating the total distance moved per each 30 s interval and the average speed. Thigmotaxis is a well-validated index of anxiety in a wide range of species, including zebrafish [1,2], which is evolutionarily conserved. Animals that are engaged in thigmotaxic behavior strongly avoid the center/innermost of an area and stay or move in close proximity to the boundaries of a novel environment, for instance the walls. Thigmotaxis is believed to be adaptive in nature and meant to facilitate the search for shelter, protection, and/or escape routes [43,44]. The percentage of time each fish spent, or total distance moved in the outer zone defined the index of thigmotaxis [1,2]. The open field tank was divided into two equal zones. The outer zone was defined as the peripheral zone of the tank, whose area was half the area of the entire tank. Time and distance were both indicators of measurement and evaluations of thigmotaxis. The test was conducted three times.
$$\mathrm{Thigmotaxis}=\frac{\mathrm{Time}/\mathrm{Distance}\;\mathrm{spent}\;\mathrm{in}\;\mathrm{the}\;\mathrm{peripheral}\;\mathrm{zone}}{\mathrm{Time}/\mathrm{Distance}\;\mathrm{spent}\;\mathrm{in}\;\mathrm{the}\;\mathrm{whole}\;\mathrm{zone}}$$

#### 4.5.3. Juvenile Social Preference Behavior

The juvenile (1 mpf and 1.5 mpf) social preference test was adapted from a previous experimental paradigm [45] and conducted using the ZebraLab behavior monitoring station (Viewpoint Life Sciences, Lyon, France). As shown in Figure 5A, fish were acclimated in 6-well plates with ample water for free swimming. The subject fish (WT or *brsk2ab*^−/−^) were placed in the middle wells of each row, with a conspecific WT fish of similar age and size in one side well and the corresponding opposite well left empty. The test comprised a baseline phase and a post-baseline phase. During the baseline phase, removable opaque partitions separated each well to prevent interference. In the post-baseline phase, these partitions were removed, allowing the subject fish to observe its corresponding conspecific fish [45]. Video recordings of each phase consisted of a 5-min habituation period followed by a 10-min experimental period. Data were collected every 30 s, capturing the time and distance the subject fish spent in the total zone and the quarter zones (empty zone and conspecific zone) near its side wells. The social preference index (SPI) of baseline and post-baseline phases were both analyzed by the distance and time the fish moved in the conspecific zone and the empty zone (showed in the following formula) [45], and the data of distance and time were obtained using the ZebraLab behavior monitoring station (Viewpoint Life Sciences, Lyon, France). Changes in SPI between the baseline and post-baseline phases were also analyzed.
$$SPI=\frac{\mathrm{Time}/\mathrm{Distance}\;\mathrm{spent}\;\mathrm{in}\;\mathrm{conspecific}\;\mathrm{sector}–\mathrm{Time}/\mathrm{Distance}\;\mathrm{spent}\;\mathrm{in}\;\mathrm{empty}\;\mathrm{sector}}{\mathrm{Time}/\mathrm{Distance}\;\mathrm{spent}\;\mathrm{in}\;\mathrm{both}\;\mathrm{sectors}}$$

Data were collected every 30 s, capturing the time and distance the subject fish spent in the total zone and the quarter zones near its side wells. The social preference index (SPI) was calculated as the proportion of time/distance spent in the conspecific zone to the total zone, providing insight into social behavior. Changes in SPI between the baseline and post-baseline phases were also analyzed.

#### 4.5.4. Adult Zebrafish Social Preference Test

For adult social preference experiments, we utilized a mating tank (21 cm × 11 cm × 7.5 cm) separated into two parts by a transparent divider (Figure 5B). The experiment was performed at both 3 mpf and 4 mpf. For each assay, one subject zebrafish (WT or *brsk2ab*^−/−^) was placed in one part, while six male WT conspecifics of similar age and size were placed in the other part. The tests lasted 15 min, consisting of a 5-min habituation period and a 10-min observation period. We recorded time and distance measurements every 30 s using Zebralab software (ViewPoint Life Sciences, Lyon, France). For the purposes of data analysis, the subject’s zone was divided into two sections: the conspecific section (nearest the 6 conspecific zebrafish) and the empty section (farther away from them). We calculated the social preference index (SPI) by determining the proportion of time/distance that the subject spent in the conspecific section compared to the total zone [7,46].
$$SPI=\frac{\mathrm{Time}/\mathrm{Distance}\;\mathrm{in}\;\mathrm{conspecific}\;\mathrm{sector}\;–\mathrm{Time}/\mathrm{Distance}\;\mathrm{in}\;\mathrm{empty}\;\mathrm{sector}}{\mathrm{Time}/\mathrm{Distance}\;\mathrm{in}\;\mathrm{total}\;\mathrm{sectors}}$$

#### 4.5.5. Shoaling Behavior Test

Shoaling is a social behavior observed in zebrafish, wherein a group of them swim closely together in an enclosure (Figure 5C). To comprehensively assess the social behavior of the *brsk2ab*^−/−^ zebrafish, six fish of the same genotype (WT or *brsk2ab*^−/−^) at 3 mpf and 4 mpf were introduced into a novel tank apparatus (30 cm × 30 cm × 30 cm). After a 5-min acclimatization period, the swimming zebrafish were recorded for an additional 10 min for assessment, and the videos were subsequently analyzed using Zebralab software (ViewPoint Life Sciences, Lyon, France). The shoaling assessment was performed by measuring the inter-fish distance that indicated the average of all distances between each zebrafish in a group, recorded by Zebralab software (ViewPoint Life Sciences, Lyon, France). Viewpoint software is a commercial software designed to analyze zebrafish behavior experiments, which enables an automated quantification of the average and median distances between any two fish in the group [47,48].

#### 4.5.6. Statistical Analysis

The statistical analysis was performed using GraphPad Prism 8. Data are presented as the mean ± standard error (SEM). To test the normality of data distribution, we used the Kolmogorov-Smirnov evaluation before conducting the statistical analysis. A Student’s *t*-test or one-way ANOVA with post hoc Turkey’s test was used to analyze continuous variables. Due to multiple comparisons of the same data set (i.e., the same individuals were used to assess three behavioral endpoints, distances moved, times active, and swimming speeds), the results were Bonferroni corrected to avoid any Type I errors. Therefore, the significance was assigned at *p* < 0.017 (0.05/3), *p* < 0.003 (0.01/3), and *p* < 0.0003 (0.001/3).

## Figures and Tables

**Figure 1 ijms-24-16506-f001:**
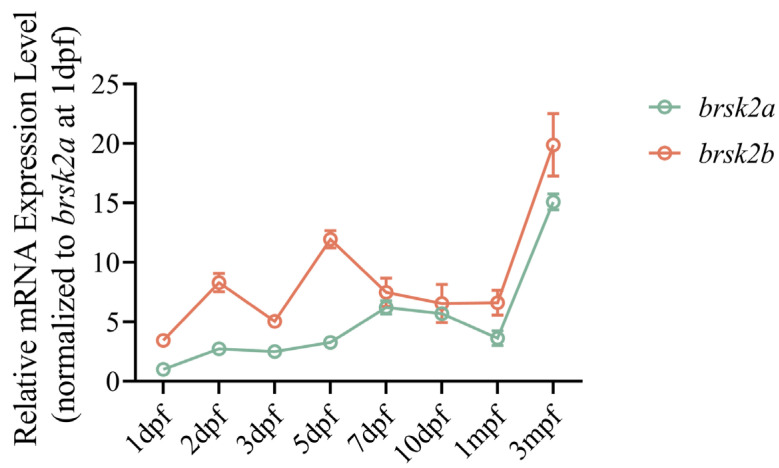
Temporal pattern of mRNA expression of zebrafish *brsk2a* and *brsk2b* at 8 stages. mRNA was extracted from whole embryos at 1 dpf, 2 dpf, 3 dpf, and brain tissues at 5 dpf, 7 dpf, 10 dpf, 1 mpf and 3 mpf. Data are shown as mean ± SEM, *n* = 3 for each genotype.

**Figure 2 ijms-24-16506-f002:**
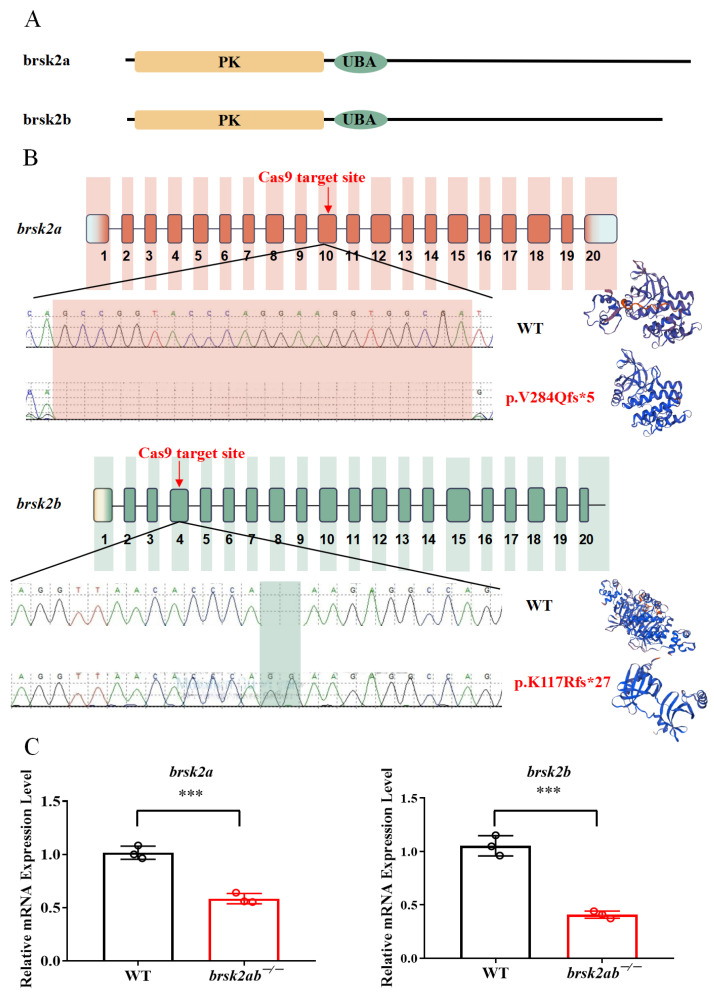
Generation of *brsk2ab* mutant zebrafish. (**A**) structure modelling of zebrafish *brsk2a* and *brsk2b* protein. PK represents protein kinase domain; UBA represents ubiquitin-associated domain. (**B**) Two CRISPR/Cas9 guide RNAs were designed to target exon 10 in *brsk2a* and exon 4 in *brsk2b*, respectively. The mutations of *brsk2a* and *brsk2b* were verified by Sanger sequencing, and computational modelling showed the proteins were truncated through SWISS-MODEL tool (https://swissmodel.expasy.org/interactive, accessed on 12 July 2023). (**C**) To validate the knockout efficiency, total mRNA was extracted at 3 mpf followed by reverse transcription, and qRT-PCR was conducted to validate the expression level of *brsk2a*, and *brsk2b* was decreased in *brsk2ab*^−/−^ zebrafish (*n* = 3 for each genotype, *brsk2a*, *p* = 0.0007; *brsk2b*, *p* = 0.0004). Data are shown as mean ± SEM, *** *p* < 0.001.

**Figure 3 ijms-24-16506-f003:**
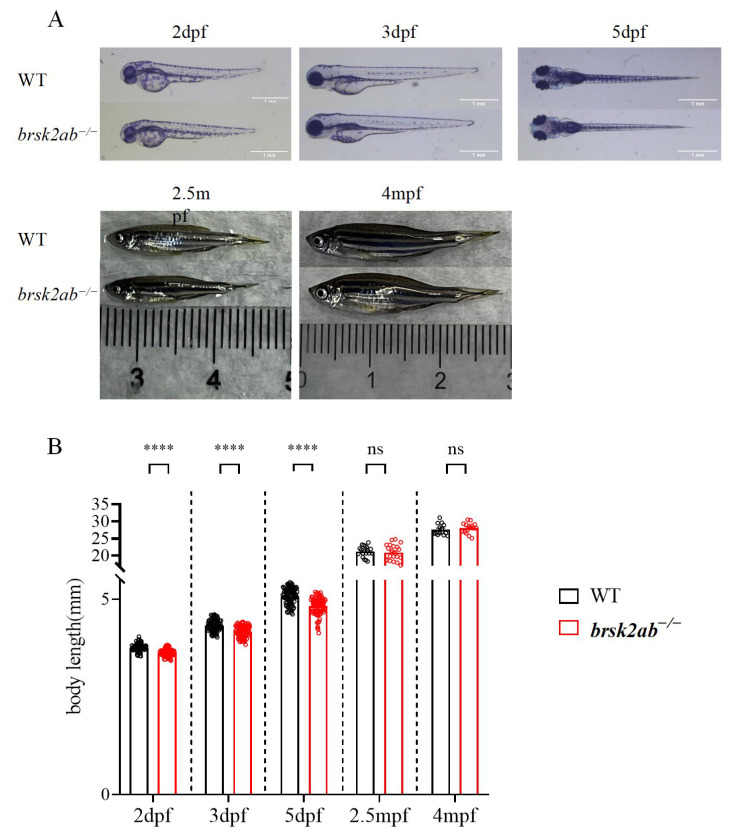
*brsk2ab*^−/−^ zebrafish showed a catch-up growth pattern. (**A**) Representative pictures of measured zebrafish. (**B**) The body lengths were measured at 2 dpf (WT: *brsk2ab*^−/−^, *n* = 120: 131), 3 dpf (*n* = 105: 105), 5 dpf (*n* = 100: 89), 2.5 mpf (*n* = 19: 21), and 4 mpf (*n* = 14: 16). The *brsk2ab*^−/−^ larvae have shorter bodies than WT larvae at 2 dpf (WT vs. *brsk2ab*^−/−^, 3.760 ± 0.09 vs. 3.623 ± 0.09; *p* < 0.0001), 3 dpf (4.329 ± 0.14 vs. 4.170 ± 0.15; *p* < 0.0001), and 5 dpf (5.071 ± 0.21 vs. 4.819 ± 0.22; *p* < 0.0001), but there was no difference in body length at 2.5 mpf (21.19 ± 1.58 vs. 20.90 ± 2.31; *p* = 0.646) and 4 mpf (27.57 ± 1.66 vs. 28.04 ± 1.53; *p* = 0.425). Data are shown as mean ± SEM, **** *p* < 0.0001.

**Figure 4 ijms-24-16506-f004:**
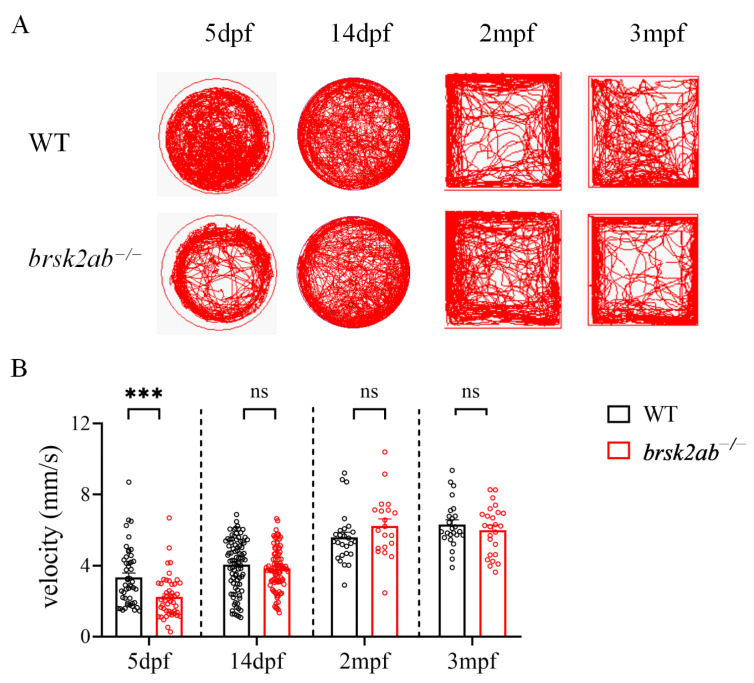
The locomotor activity of *brsk2ab*^−/−^ zebrafish in different stages. (**A**) The trajectory of locomotion in WT and *brsk2ab*^−/−^ zebrafish in different time points. (**B**) The *brsk2ab*^−/−^ zebrafish presented significantly reduced swimming velocity at 5 dpf (WT vs. *brsk2ab*^−/−^, 3.342 ± 1.63 vs. 2.229 ± 1.18; *p* = 0.0002). While no significant difference was observed at 14 dpf (4.046 ± 1.50 vs. 3.826 ± 1.18; *p* = 0.2753) and later stages (2 mpf: 5.589 ± 1.47 vs. 6.234 ± 1.75; *p* = 0.1087; 3 mpf: 6.309 ± 1.31 vs. 5.998 ± 1.37; *p* = 0.2745). At 5 dpf: *n* = 48: 48; at 14 dpf: *n* = 89: 91; at 2 mpf: *n* = 26: 20; and at 3 mpf: *n* = 25: 25. Data are shown as mean ± SEM, *** *p* < 0.0003.

**Figure 5 ijms-24-16506-f005:**
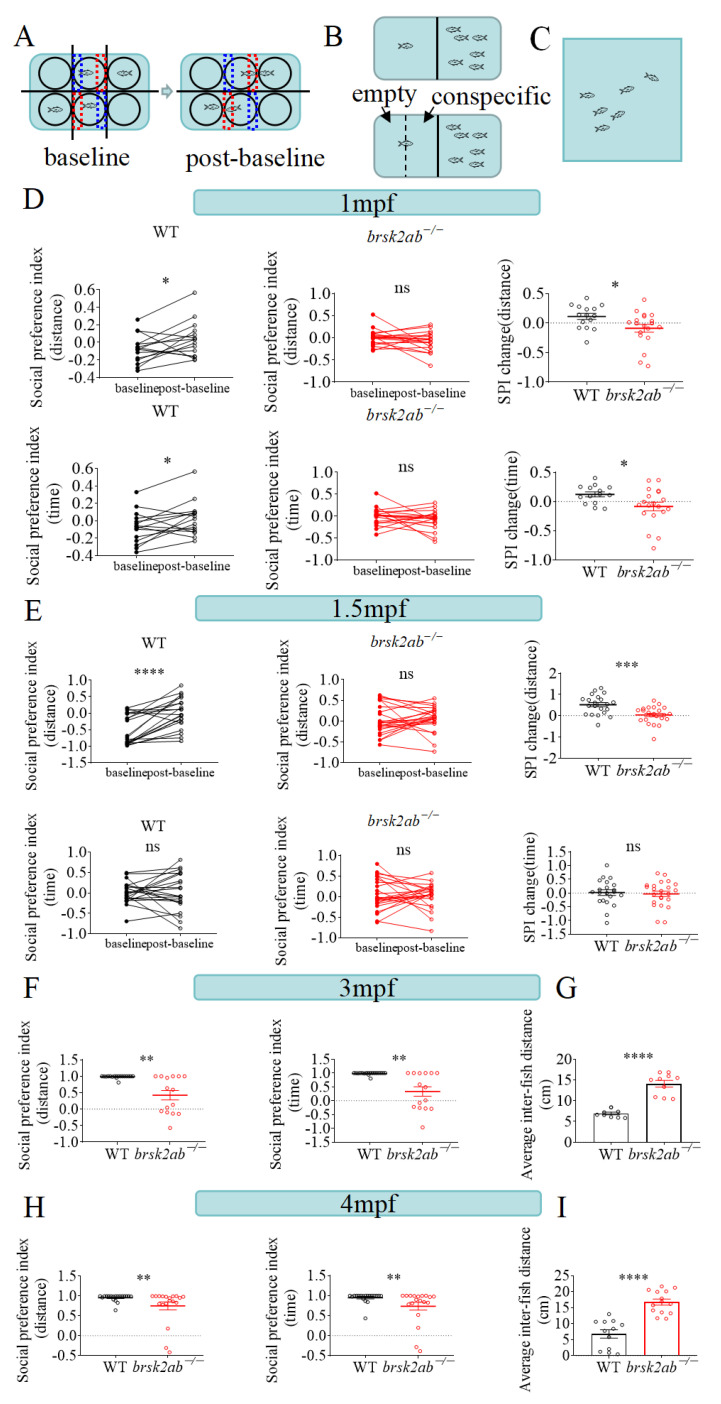
*brsk2ab*^−/−^ zebrafish displayed gradually severe social defects as they grew up. The experimental paradigm of social preference tests at juvenile stage (**A**) and adult stage (**B**). (**C**) The experimental paradigm of shoaling tests at adult stage. (**D**) The *brsk2ab*^−/−^ zebrafish showed specious defects in social preference at 1 mpf and the SPI change was smaller than that of WT zebrafish (WT vs. *brsk2ab*^−/−^, *n* = 15: 19; distance: 0.1116 ± 0.20 vs. −0.08843 ± 0.29, *p* = 0.0298; time: 0.1228 ± 0.16 vs. −0.0827 ± 0.32, *p* = 0.0327). (**E**) The discrepancy of SPI change was obvious at 1.5 mpf (WT vs. *brsk2ab*^−/−^, *n* = 21: 23; distance: 0.5166 ± 0.46 vs. 0.03154 ± 0.40, *p* = 0.0005; time: 0.0160 ± 0.48 vs. −0.0307 ± 0.47, *p* = 0.7444). (**F**,**H**) The experimental paradigm of social preference tests in adulthood. The SPI of adult *brsk2ab*^−/−^ zebrafish was significantly decreased compared to WT fish both in distance and time ratios at 3 mpf (WT vs. *brsk2ab*^−/−^, *n* = 18: 15, distance: 0.9858 ± 0.04 vs. 0.4247 ± 0.56, *p* = 0.0022; time: 0.9846 ± 0.05 vs. 0.3328 ± 0.66, 0.0029) and 4 mpf (WT vs. *brsk2ab*^−/−^, *n* = 19: 19, distance: 0.9589 ± 0.09 vs. 0.7523 ± 0.43, *p* = 0.0072; time: 0.9486 ± 0.13 vs. 0.7361 ± 0.43, *p* = 0.0044). (**G**,**I**) The shoaling behavior tests exhibited the average inter-individual distance of *brsk2ab*^−/−^ zebrafish was smaller than that of WT fish at 3 mpf (WT vs. *brsk2ab*^−/−^, *n* = 9: 10, 6.935 ± 0.99 vs. 14.07 ± 2.58, *p* < 0.0001) and 4 mpf (WT vs. *brsk2ab*^−/−^, *n* = 12: 14, 6.785 ± 4.51 vs. 16.75 ± 3.56, *p* < 0.0001). Data for each genotype are presented as mean ± SEM. * *p* < 0.05, ** *p* < 0.01, *** *p* < 0.001,**** *p* < 0.0001.

**Figure 6 ijms-24-16506-f006:**
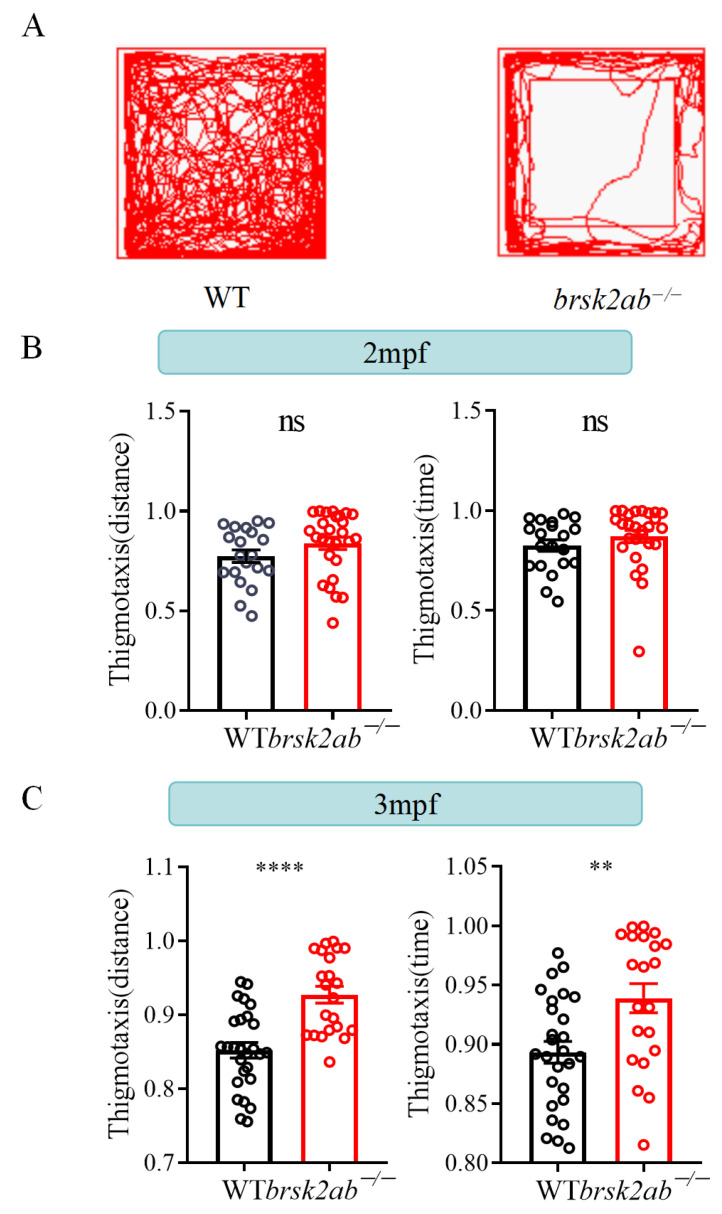
*brsk2ab*^−/−^ zebrafish displayed escalating anxiety-like behaviors. (**A**) The analytical paradigm showed that the open field tank was divided into two equal zones, and the thigmotaxis was calculated using the time and distance ratio that fish spent in the peripheral zone. (**B**) There was no significant difference in the thigmotaxis index between WT and *brska2b*^−^*^/^*^−^ zebrafish at 2 mpf (WT vs. *brsk2ab*^−/−^, *n* = 20: 27, distance: 0.7737 ± 0.14 vs. 0.8376 ± 0.16, *p* = 0.1622; time: 0.8245 ± 0.13 vs. 0.8712 ± 0.15, *p* = 0.2225). (**C**) The thigmotaxis of *brsk2ab*^−/−^ zebrafish was observed to have increased more than that of WT zebrafish at 3 mpf (WT vs. *brsk2ab*^−/−^, *n* = 27: 22, distance: 0.8522 ± 0.05 vs. 0.9274 ± 0.05, *p* < 0.0001; time: 0.8934 ± 0.05 vs. 0.9389 ± 0.06, *p* = 0.0038). Data are shown as mean ± SEM s, ** *p* < 0.01, **** *p* < 0.0001.

## Data Availability

The data that support the findings of this study are available from the corresponding authors upon request.

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
