# Peer review of "Dynamic Regulation of brsk2 in the Social and Motor Development of Zebrafish: A Developmental Behavior Analysis"

_ijms, 2023, doi:10.3390/ijms242216506_

Round 1
Reviewer 1 Report
Comments and Suggestions for Authors
Comments to authors:
Manuscript entitled “Dynamic regulation of brsk2 in the social and motor development of zebrafish: A developmental behavior analysis” by Jingxin et al. has been reviewed.
The manuscript provides essential information. However, authors should address the following points:
1. Page 2, Line 67. Please replace “BRSK2 mutant patients” by a more appropriate term such as “mutations in the BRSK2 gene …” Patients cannot be mutants.
2. Line 87, please clearly state that at 1 to 3 dpf whole embryo were used and not whole brain, and explain why.
3. Lines 95-96. Authors need to discuss reason why there is a decline in expression post 7 dpf.
4. Figure 1. Please add individual values as supplemental information.
5. Page 4. Line 117. Please add a link for Swiss-model.
6. Figure 3. Please add the pictures of the measured zebrafish (representative pictures).
7. Figure 4. It should be velocity, as the unit is in mm/s. Distance should be in mm. Please also show representative pictures of the behavioral traces of both groups.
8. Figure 5. From D to G. How were the graph constructed is unclear. Similarly, please add as supplementary information, the individual values.
9. Were the same fish used for social preference tests and shoaling test?
10. Figure 6. How is thigmotaxis evaluated? What is the index of thigmotaxis? Authors should describe appropriately, as these graphs are unclear. Page 220, …by the time and distance ratio? Figure 6 shows thigmotaxis (time) and (distance). What are the differences? Please write a formula for calculation or explain better.
11. Line 221, …was no difference… there is a grammatical error in the sentence.
12. Line 228: is it the first brsk2-deficient zebrafish?
13. Line 262-263. Please consider updating this interesting review: DOI: 10.3390/ijms23147550
14. Line 275-277 is repeated.
15. Line 340-344 is unclear. Please describe properly what was crossed and how. Supplementary Table S1 does not show crossing information.
16. Line 370. Please give references for the VideoTrack software.
17. Line 395. Please provide information about the removable opaque partition used for 6-well to allow other researchers to reproduce.
18. Line 402-404 is unclear. Please use clear formulas. How was the post-baseline analyzed? Please add references also.
19. Same for lines 414-416.
20. Line 424 is unclear. Please explain how it was calculated.
21. Methods for measuring anxiety are missing in the manuscript.
22. Figure S1: what is the difference between total distance and average distance? Please note you mention average distance for 5 and 14dpf and then total distance for 2 and 3 mpf. Please describe the meaning and show representative tracking traces.
Comments on the Quality of English LanguageMinor editing required
Author Response
Author's Reply to the Review Report (Reviewer1)
The manuscript provides essential information. However, authors should address the following points:
Comments 1: Page 2, Line 67. Please replace “BRSK2 mutant patients” by a more appropriate term such as “mutations in the BRSK2 gene …” Patients cannot be mutants.
Response 1: We agree with this comment and have revised this in the updated manuscript.
“Furthermore, research indicates that patients with mutations in the BRSK2 gene exhibited deficits in social behavior and communication skills, pivotal aspects of human interaction that significantly influence different facets of life.” (Page 2, Line 67)
Comments 2: Line 87, please clearly state that at 1 to 3 dpf whole embryo were used and not whole brain, and explain why.
Response 2: Thank you for pointing this out. Indeed, the best approach is to obtain zebrafish brain tissues from 1 to 3 dpf for testing. However, due to the small sizes of zebrafish during these periods, it is difficult to obtain complete brain tissues. Therefore, in the first three days, we chose to use the entire embryos for experimentation.
Comments 3: Lines 95-96. Authors need to discuss reason why there is a decline in expression post 7 dpf.
Response 3: Thank you for reviewer’s insightful comments. We have added the discussion to emphasize this point.
“For example, in this study, the expression levels of brsk2a and brsk2b increased in the first 5-7 days and then showed a slight decreasing trend, but they subsequently exhibited upward trend towards juvenile and adulthood. The specific mechanisms behind this phenomenon were unclear. Here are some possible explanations. During early embryonic stages, various organs and tissues are rapidly developing and differentiating. brsk2 may be involved in specific growth signaling pathways or regulatory mechanisms, including structural and functional maturation of synapses[15], leading to an increase in its expression during this stage. As organ development completes and enters the stage of functional maturity, the expression level of brsk2 may gradually decrease. During individual development, different regulatory mechanisms and molecular signaling pathways may undergo changes. These changes may result in a re-increase in the expression level of brsk2 in juvenile and adulthood, possibly due to new regulatory factors or signaling pathways beginning to influence its expression.” (Page 11, Line 257-270)
Comments 4: Figure 1. Please add individual values as supplemental information.
Response 4: Agree. We have added the individual values in the Supplementary Figure S1.
Comments 5: Page 4. Line 117. Please add a link for Swiss-model.
Response 5: SWISS-MODEL tool (https://swissmodel.expasy.org/interactive) (Page 4, Line 119-120)
Comments 6: Figure 3. Please add the pictures of the measured zebrafish (representative pictures).
Response 6: Agree. We have revised the Figure 3.
Comments 7: Figure 4. It should be velocity, as the unit is in mm/s. Distance should be in mm. Please also show representative pictures of the behavioral traces of both groups.
Response 7: Thank you for pointing this out. We agree with this comment. We have revised the Figure 4.
Comments 8: Figure 5. From D to G. How were the graph constructed is unclear. Similarly, please add as supplementary information, the individual values.
Response 8: We have revised the Figure 5. Scatter plots for each test zebrafish (individual values) have been shown including juvenile social preference test (Figure 5D-E) and adult social preference test (Figure 5F-I)
Comments 9: Were the same fish used for social preference tests and shoaling test?
Response 9: Yes, the same zebrafish were used in the adult social preference tests and shoaling tests.
Comments 10: Figure 6. How is thigmotaxis evaluated? What is the index of thigmotaxis? Authors should describe appropriately, as these graphs are unclear. Page 220, …by the time and distance ratio? Figure 6 shows thigmotaxis (time) and (distance). What are the differences? Please write a formula for calculation or explain better.
Response 10: Thank you for pointing this out. We have revised in the updated manuscript.
“Thigmotaxis is a well-validated index of anxiety in a wide range of species, including zebrafish[1,2], which is evolutionarily conserved. Animals that are engaged in thigmotaxic behavior strongly avoid the center/inner of an area and stay or move in close proximity to the boundaries of a novel environment, for instance the walls. Thigmotaxis is believed to be adaptive in nature and meant to facilitate the search for a shelter, protection and/or escape routes[43,44] . The percentage of each fish’s spent or total distance moved in the outer zone defined the index of thigmotaxis [1,2]. The open field tank was divided into two equal zones. The outer zone was defined as the peripheral zone of the tank, whose area was half the area of the entire tank. Time and distance were both indicators of measurement and evaluation of thigmotaxis. The test was conducted three times. ”(Page 14, Line 409-420)
Comments 11: Line 221, …was no difference… there is a grammatical error in the sentence.
Response 11: We agree with this comment. Therefore, we have modified the sentence.
“There was no significant difference in the thigmotaxis index between WT and brska2b−/− zebrafish at 2mpf.” (Page 10, Line 223)
Comments 12: Line 228: is it the first brsk2-deficient zebrafish?
Response 12: Yes, this is the first reported brsk2-deficient zebrafish model to date.
Comments 13: Line 262-263. Please consider updating this interesting review: DOI: 10.3390/ijms23147550
Response 13: Agree. We have added this excellent supporting evidence. (Page 11, Line 277)
Comments 14: Line 275-277 is repeated.
Response 14: Yes, we have removed this repeated sentence. (Page 11, Line 290)
Comments 15: Line 340-344 is unclear. Please describe properly what was crossed and how. Supplementary Table S1 does not show crossing information.
Response 15:
We have revised the crossing protocol in detail as following and added crossing strategy in Supplementary Figure S3:
“The F0 adult zebrafish were out-crossed with WT zebrafish for at least three generations and genotyping of each generation to obtain F4 heterozygous zebrafish. Homozygous brsk2a-/- or brsk2b-/- zebrafish were obtained through genotype identification after in-crossing the F4 heterozygous adult zebrafish. The brsk2ab-/- homozygous line was obtained by crossing homozygous brsk2a-/- and brsk2b-/ -and subsequent genotyping (Supplementary Figure S3). The genotyping PCR amplification conditions were as follows: 95â—¦C, 4 min; 35 cycles of 95â—¦C, 30 s; 58â—¦C, 30 s; 72â—¦C, 45 s; 72â—¦C, 7 min. The PCR primers for genotyping were listed in Supplementary Table S1. “(Page 13, Line 359-366)
Comments 16: Line 370. Please give references for the VideoTrack software.
Response 16: Yes, we have added the references. (Page 14, Line 394)
Comments 17: Line 395. Please provide information about the removable opaque partition used for 6-well to allow other researchers to reproduce.
Response 17: We conducted the juvenile social preference test following a previously described experimental paradigm[45] and it has already been explained in the first sentence of this paragraph (Page 14, Line 423 and Line 431).
Comments 18: Line 402-404 is unclear. Please use clear formulas. How was the post-baseline analyzed? Please add references also.
Response 18: we have added the formulas and references[45].
“Data were collected every 30 seconds, capturing the time and distance the subject fish spent in the total zone and the quarter zones (empty zone and conspecific zone) near its side wells. The social preference index (SPI) of baseline and post-baseline phases were both analyzed by the distance and time the fish moved in conspecific zone and empty zone (showed in the following formula), and the data of distance and time were obtained by the ZebraLab behavior monitoring station (Viewpoint Life Sciences). Changes in SPI between the baseline and post-baseline phases were also analyzed.” (Page 14, Line 433-441)
Comments 19: Same for lines 414-416.
Response 19: we have added the formulas and references [7,46]. The social preference was assessed by SPI and calculated as showed in Page 15, Line 460.
Comments 20: Line 424 is unclear. Please explain how it was calculated.
Response 20: The shoaling assessment was performed by measuring the inter-fish distance that indicated the average of all distances between each zebrafish in a group calculated by Viewpoint software. Viewpoint software is a commercial software designed to analyze zebrafish behavior experiments, which enable automated quantification of the average and median distances between any two fish in the group [47,48].
“The shoaling assessment was performed by measuring the inter-fish distance that indicated the average of all distances between each zebrafish in a group calculated by Viewpoint software. Viewpoint software is a commercial software designed to analyze zebrafish behavior experiments, which enable automated quantification of the average and median distances between any two fish in the group [47,48]”.
Comments 21: Methods for measuring anxiety are missing in the manuscript.
Response 21: It is included in 4.5.2. Open field test part. The index of thigmotaxis was determined anxiety levels. (Page 14, Line 409-420)
Comments 22: Figure S1: what is the difference between total distance and average distance? Please note you mention average distance for 5 and 14dpf and then total distance for 2 and 3 mpf. Please describe the meaning and show representative tracking traces.
Response 22: Both the Indicators of “total distance moved per 30 seconds” and “average speed” were calculated. And the histogram (Figure 4) was made by the average speed, which is the velocity. The Figure S1 were made using the total distance moved per 30 seconds.
We have revised the ordinate of Figure 4 and the units of the ordinate in Figure S1.
Reviewer 2 Report
Comments and Suggestions for Authors
the manuscript investigated the role of the brsk2 gene on zebrafish development through the generation of the brsk2ab defect zebrafish line. The authors designed the study nicely and the manuscript was written well.
More details need to be added to the Zebrafish maintenance and husbandry section.
why Exon 10 and Exon 4 was selected for sgRNA, why not the promoter region?
The endpoints such as speed are dependent on the time of swimming and distance moved, therefore the associated p-value needs to be corrected to avoid type I error. please see https://www.mdpi.com/1422-0067/18/2/291
Comments on the Quality of English LanguageEnglish is fine.
Author Response
Author's Reply to the Review Report (Reviewer2)
the manuscript investigated the role of the brsk2 gene on zebrafish development through the generation of the brsk2ab defect zebrafish line. The authors designed the study nicely and the manuscript was written well.
Comments 1: More details need to be added to the Zebrafish maintenance and husbandry section.
Response 1: Thank you for pointing this out. We have added more details.
“The wild-type Tu strain was used in this study. All zebrafish were raised and maintained under standard laboratory conditions of 28.5 °C under a 14-hour light/10-hour dark photoperiod in a recirculation system according to standard zebrafish breeding protocols[37]. For breeding, adult zebrafish were placed in tanks overnight with a sex ratio of 1:1 and were separated by a transparent barrier that was removed on the following morning. Freshly fertilized eggs were collected within 0.5 h of spawning and immersed in the methylene blue solution and maintained in an incubator at 28.5°C until 5 dpf. Larvae aged 5-12 dpf were nourished with paramecium and subsequently transferred to the recirculating water system at 28.5 °C .Water supplied to the system was filtered by reverse osmosis (pH 7.0–7.5). All animal experimental procedures were in compliance with local and international regulations, and approved by the institutional animal care committee of Children's Hospital of Fudan University.” (Page 12, Line 337-348)
Comments 2: Why Exon 10 and Exon 4 was selected for sgRNA, why not the promoter region?
Response 2: In the original design, we designed CRISPR/Cas9 targets at different locations in brsk2a and brsk2b. Finally, strains with higher knock-out efficiency and more harmful variants predicted by the software were selected to be retained.
Comments 3: The endpoints such as speed are dependent on the time of swimming and distance moved, therefore the associated p-value needs to be corrected to avoid type I error. please see https://www.mdpi.com/1422-0067/18/2/291
Response 3: We agree with this comment. Therefore, we have modified the statistical analysis and Figure 4 as well as figure legends.
“Due to multiple comparisons of the same data set (i.e., the same individuals were used to assess three behavioral endpoints, distance moved, time active, and swimming speed), the results were Bonferroni corrected to avoid Type I errors. Therefore, significance was assigned at p < 0.017 (0.05/3), p < 0.003(0.01/3) and p < 0.0003(0.001/3).” (Page 15, Line 479-482)